# Prevalence of visual impairment due to refractive error among children and adolescents in Ethiopia: A systematic review and meta-analysis

Daniel Atlaw[1]*, Zerihun Shiferaw[1], Biniyam Sahiledengele[2], Sisay Degno[3], Ayele Mamo[4], Demisu Zenbaba[2], Habtamu Gezahegn[1], Fikreab Desta[2], Wogene Negash[5], Tesfaye Assefa[5], Mujib Abdela[6], Abbul Hasano[1], Gashaw Walle[7], Chala Kene[6], Degefa Gomora[6], Vijay Kumar Chattu[8,9,10]

1 Department of Biomedical Sciences, Madda Walabu University Goba Referral Hospital, Bale-Goba, Ethiopia, 2 Department of Public Health, Madda Walabu University Goba Referral Hospital, Bale-Goba, Ethiopia, 3 Department of Public Health, Madda Walabu University Shashemene Campus, Shashemene, Ethiopia, 4 Department of Pharmacy, Madda Walabu University Goba Referral Hospital, Bale-Goba, Ethiopia, 5 Department of Nursing, Madda Walabu University Goba Referral Hospital, Bale-Goba, Ethiopia, 6 Department of Midwifery, Madda Walabu University Goba Referral Hospital, Bale-Goba, Ethiopia, 7 Department of Biomedical Sciences, Debre Tabor University, Debre Tabor, Ethiopia, 8 Department of Occupational Science and Occupational Therapy, Temetry Faculty of Medicine, University of Toronto, Toronto, Ontario, Canada, 9 Center for Transdisciplinary Research, Saveetha Institute of Medical and Technological Sciences, Saveetha University, Chennai, India, 10 Department of Community Medicine, Faculty of Medicine, Datta Meghe Institute of Medical Sciences, Wardha, India

* danielatmwu@gmail.com

## Abstract

### Introduction

Globally, the prevalence of refractive error was 12%, and visual impairment due to refractive error was 2.1%. In sub-Saharan Africa, the prevalence of refractive error and visual impairment due to refractive error was 12.6% and 3.4%, respectively. In Ethiopia, the prevalence of visual impairment due to refractive error varies from 2.5% in the Gurage zone to 12.3% in Hawassa city. Hence, this Meta-analysis aimed to summarize the pooled prevalence of visual impairment due to refractive error in Ethiopia.

### Methods

A systematic search of the literature was conducted by the authors to identify all relevant primary studies. All articles on the prevalence of visual impairment due to refractive error in Ethiopia were identified through a literature search. The databases used to search for studies were PubMed, Science Direct, POPLINE, HENARI, Google Scholar, and grey literature was searched on Google until December 15, 2021. In this meta-analysis, the presence of publication bias was evaluated using funnel plots and Begg's tests at a significance level of less than 0.05. The sensitivity analysis was conducted to check for a single study's effect on the overall prevalence of refractive error.

**Data Availability Statement:** All relevant data are within the paper and its Supporting information files.

**Funding:** The authors received no specific funding for this work.

**Competing interests:** The authors have declared that no competing interests exist.

## Result

About 1664 studies were retrieved from initial electronic searches using international databases and google searches. A total number of 20,088 children and adolescents were included in this meta-analysis. The pooled prevalence of visual impairment due to refractive error in Ethiopia using the random effects model was estimated to be 6% (95% CI, 5–7) with a significant level of heterogeneity ($I^2 = 94.4\%$; $p < 0.001$). The pooled prevalence of visual impairment due to refractive was analyzed by subtypes, and pooled prevalence was estimated to be 4%, 5.2%, and 1% for myopia, hyperopia, and astigmatism, respectively.

## Conclusion

The pooled prevalence of visual impairment due to refractive error was high in Ethiopia. About one in twenty-five Ethiopian children and adolescents are affected by visual impairment due to myopia.

## Introduction

Refractive error (RE) occurs when the eye's optical system fails to adjust to bring parallel beams of light into proper focus on the fovea [1, 2]. Refractive error includes myopia, hyperopia, astigmatism, and presbyopia. It is the most common cause of reduced visual impairment [3]. More than 43% of visual impairment is caused by uncorrected RE [4, 5], which is the second leading cause of blindness [1].

Globally the prevalence of RE was 12% [4], and visual impairment due to RE was 2.1% [6]. A review and meta-analysis conducted in India revealed the prevalence of visual impairment due to RE to be 10.8% [7]. In sub-Saharan Africa, the prevalence of refractive error and visual impairment due to RE was 12.6% and 3.4%, respectively, in 2015 [5]. In Ethiopia, the prevalence of visual impairment due to refractive error varies from 2.5% in the Gurage zone [8] to 12.3% in Hawassa city [9].

The global initiative to eliminate avoidable blindness by 2020 (VISION 2020: the Right to Sight) has included RE as one of the five priority eye diseases, following epidemiological studies highlighting the escalating estimates of RE prevalence [10, 11]. Over 80% of visual impairments are preventable; therefore, reducing avoidable visual impairment remains an important international public health goal [12].

Visual impairment impacts economic growth, home care, and rehabilitation that account for a fraction of the annual monetary cost and loss of quality-adjusted life-years that can be averted by fifteen percent if the refractive error is corrected [13–15]. In addition to the direct impact on quality-of-life visual impairment also affects the educational system by impairing reading speed, accuracy, and fluency [16].

Even if many studies were conducted in different parts of Ethiopia on visual impairment due to refractive error [2, 8, 9, 17–35], their results were varying and uneven. For instance, 1.9% in Jima [30], 11% in Gondar [20], and 7.2% in Wolkite [33]. Hence, this meta-analysis aimed to summarize Ethiopia's pooled prevalence of visual impairment due to refractive error.

The findings from this meta-analysis will benefit health planners and concerned bodies to act on uncorrected refractive error, thereby reducing the prevalence of visual impairment due to modifiable refractive error in Ethiopia. The findings can also be used by various governmental and non-governmental organizations working in eye care to prioritize the problem of vision impairment in Ethiopia.

## Methods

### Study design and reporting

A review and meta-analysis were conducted to estimate the pooled prevalence of visual impairment due to refractive error in Ethiopia among children and adolescents. This review and meta-analysis were conducted according to Preferred Reporting Items for Systematic Reviews and Meta-Analysis (PRISMA) guidelines [S1 Checklist].

### Eligibility criteria

Individual studies done in Ethiopia to determine the prevalence of visual impairment due to refractive error among children following criteria were included in the meta-analysis.

**Population.** All studies that had reported the prevalence of visual impairment due to refractive error among children.

**Study designs.** Observational studies reporting the prevalence of visual impairment due to refractive error among children were eligible for this systematic review and meta-analysis.

**Publication status.** Published and unpublished articles were considered.

**Year of publication.** All publications reported until December 15, 2021, were considered.

**Exclusion criteria.** Studies that reported visual impairment but did not have a separate outcome for refractive error were excluded from the study.

### The outcome of this systematic review and meta-analysis

**Visual impairment due to refractive error.** Visually impaired children due to amblyopia, hyperopia, and astigmatism. For visual acuity impairment, the participants in the primary study were visually impaired when their VA was 6/12 or worse in "either eye."

### Search strategy

A systematic search of the literature was conducted by the authors to identify all relevant primary Studies. All articles on the prevalence of visual impairment due to refractive error in Ethiopia were identified through a literature search. The databases used to search for studies were PubMed, Science Direct, POPLINE, HENARI, Google Scholar, and grey literature was searched on Google until December 15, 2021. The following key search terms and Medical Subject Headings [MeSH] were used "refractive" OR "amblyopia" OR "astigmatism" OR "hyperopia" AND "Children" AND "Ethiopia [MeSH]" were used separately or in combination with the Boolean operator's terms "AND" and "OR" [S1 File]. Moreover, the reference lists of the retrieved studies were also scanned to access additional articles and screened against our eligibility criteria. Further Ethiopian university websites were used to search for unpublished articles.

### Study selection

In this meta-analysis, all the searched articles were exported into the EndNote version X8 software, and subsequently, the duplicate articles were removed. Screening of retrieved article titles, abstracts, and the full text was conducted independently by two review authors (DA & ZS) based on the eligibility criteria. Afterward, full-text articles were retrieved and appraised to approve eligibility. Finally, the screened articles were compiled together by the two investigators.

### Risk of bias assessment

The qualities of the included studies were assessed, and the risks for biases were refereed using the Joanna Briggs Institute (JBI) quality assessment tool for the prevalence studies. Two

reviewers (DA and ZS) assessed the quality of included studies independently, and a discrepancy between the two reviewers was resolved with discussion. The evaluation tool comprises nine parameters: (1) appropriate sampling frame, (2) correct sampling technique, (3) acceptable sample size, (4) study subject and location explanation, (5) appropriate data investigation, (6) use of valid methods for the identified conditions, (7) valid measurement for all participants, (8) using appropriate statistical analysis, and (9) adequate response rate [36]. Failure to satisfy each parameter was scored as 1 if not 0. When the information provided was not satisfactory to assist in deciding on a specific item, we agreed to grade that item as 1. The risks for biases were classified as either low (total score, 0 to 2), moderate (total score, 3 or 4), or high (total score, 5 to 9) [S1 Table].

## Data extraction

The selected articles were carefully reviewed, and the required information for the meta-analysis was extracted and summarized using an extraction table in Microsoft Office Excel software. The data extraction was conducted by the two authors (DA and ZS) based on prespecified headings that are agreed on by discussion. The data extraction tool consists of the name of the author (s), year of publication, year of study, professionals conducting the visual examination, region, study design, study setting, study population, sample size, prevalence, and risk of bias.

## Statistical methods and analysis

The extracted data were imported into STATA version 14 software for statistical analysis. The heterogeneity among all included studies was assessed by $I^2$ statistics and the Cochran Q test. In this meta-analysis, the tests indicate the presence of significant heterogeneity among included studies ($I^2$ = 98.5%, P-value < 0.001). The random-effects model was used to analyze the data because the heterogeneity is high, and 22 studies were included in this meta-analysis [37, 38]. The pooled prevalence of visual impairment due to refractive error along their corresponding 95% CI was presented using a forest plot. Subgroup analyses were conducted using the region of the primary study, year of publication, year of study, study setting, sample size, eye examiners' profession, and risk of bias. Meta-regression analysis evaluated the association between the prevalence of refractive error and publication year, study year, and sample size among the included studies. In this meta-analysis, the presence of publication bias was evaluated using funnel plots [39] and Begg's tests at a significance level of less than 0.05. Begg's test was selected because of the dichotomous nature of the outcome [40]. The sensitivity analysis was conducted to check for a single study's effect on the overall prevalence of refractive error.

## Results

### Description of included studies

About 1664 studies were retrieved from initial electronic searches using international databases and google searches. The database included PubMed (n = 16), HENARI (n = 266), POPLINE(n = 235) Science Direct (n = 46), Scopus(n = 157), Google Scholar (n = 942), and Addis Ababa University repository (n = 2) studies. Of these, 1470 duplicates were removed, the remaining 194 articles were screened by title, and 131 articles were excluded after reading their titles. Sixty-three full-text articles remained and were further assessed for their eligibility. Finally, based on the pre-defined inclusion and exclusion criteria, a total of 22 articles were included in the meta-analysis, and data were extracted for the final analysis [Fig 1].

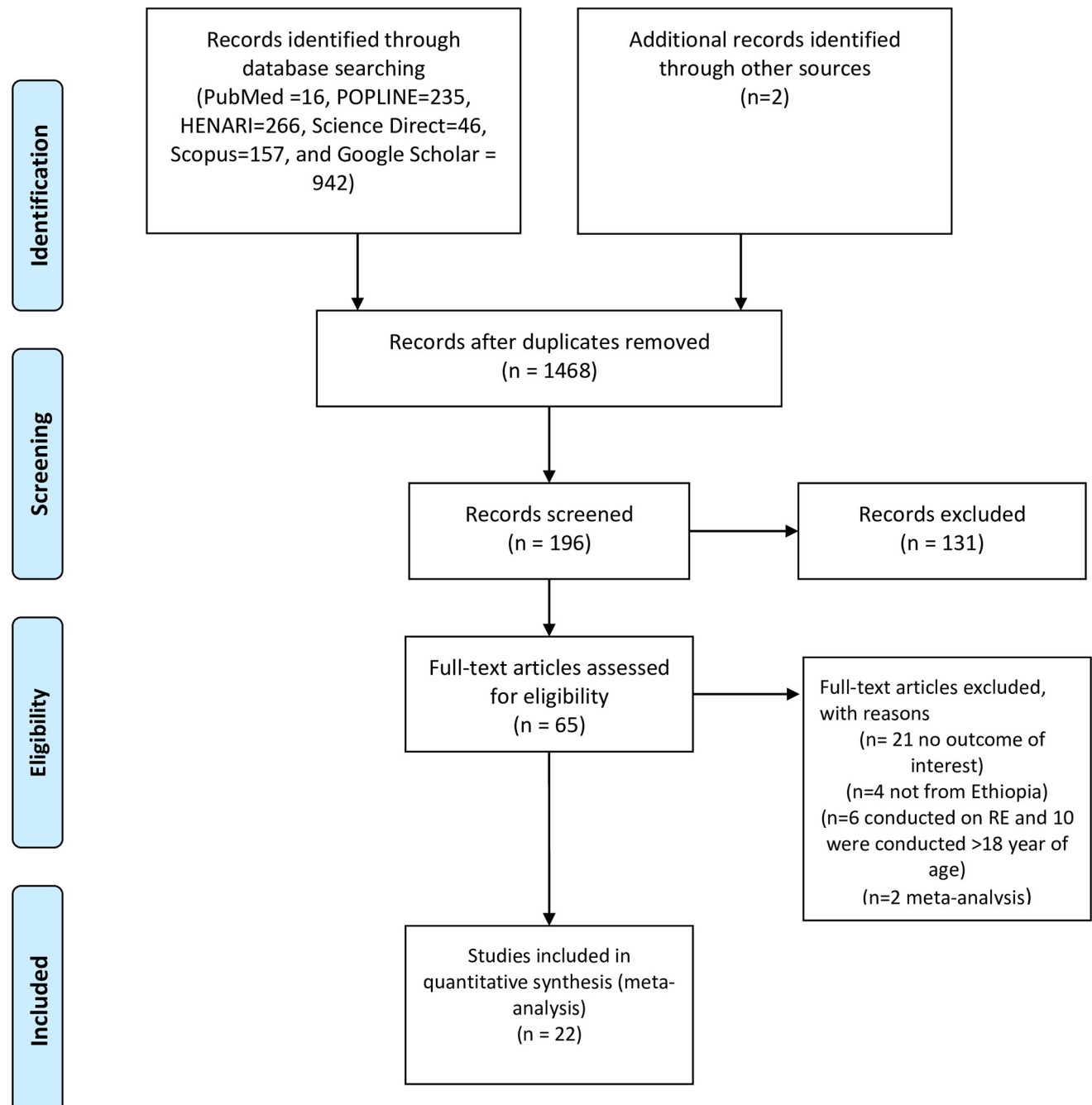

**Fig 1. Flow diagram of systemic review and meta-analysis conducted on visual impairment due to refractive error in Ethiopia, 2021.**

## Characteristics of the included studies

A total number of 20,088 children and adolescents were included in this meta-analysis. All included studies are cross-sectional in design. The latest article was conducted in 2020, and the earliest study was concluded in 2000. Depending on sample size, sixteen studies have a sample

**Table 1. Characteristics of included studies for the pooled prevalence of visual impairment due to a refractive error in Ethiopia, 2021.**

| Author name | Year of publication | Region | Study design | Sample size | Risk of bias | Prevalence of RE | Study setting | Diagnosed by | year of study |
|---|---|---|---|---|---|---|---|---|---|
| Zelalemet al.l [17] | 2019 | Amhara | Cs | 875 | Low | 0.038 | Community-based | Optometrist | 2016 |
| Sewunet et al. [2] | 2014 | Amhara | Cs | 420 | Low | 0.088 | School-based | Optometrist | 2013 |
| Kassa and Alene [18] | 2004 | Amhara | Cs | 1134 | Low | 0.076 | School-based | Not mentioned | 2000 |
| Weldeamanuel et al. [8] | 2020 | SNNP | Cs | 1064 | Low | 0.025 | School-based | Ophthalmologist | 2017 |
| Alem and Gebru [9] | 2021 | SNNP | Cs | 554 | low | 0.123 | School-based | Optometrist | 2016 |
| Darge et al. [19] | 2017 | Addis Ababa | Cs | 378 | medium | 0.05 | School-based | Ophthalmologist | 2015 |
| Ferede et al. [20] | 2020 | Amhara | Cs | 1289 | medium | 0.026 | School-based | Ophthalmologist | 2016 |
| Tegegne et al. [21] | 2021 | Amhara | Cs | 601 | medium | 0.085 | School-based | Optometrist | 2019 |
| Yared et al. [22] | 2012 | Amhara | Cs | 1852 | low | 0.049 | School-based | Optometrist | 2010 |
| Bezabih et al. [23] | 2017 | Addis Ababa | Cs | 718 | medium | 0.036 | School-based | Optometrist | 2016 |
| Hailuet al.l [24] | 2020 | Addis Ababa | Cs | 773 | medium | 0.041 | School-based | Ophthalmologist | 2019 |
| Mehari and Yimer [25] | 2012 | SNNP | Cs | 4238 | medium | 0.063 | School-based | Optometrist | 2010 |
| Kedir and Girma [27] | 2010 | SNNP | Cs | 570 | low | 0.035 | Community-based | Not mentioned | 2018 |
| Dhanesha et al. [28] | 2018 | Tigray | Cs | 1137 | low | 0.058 | School-based | Not mentioned | 2012 |
| Mehari [35] | 2014 | SNNP | Cs | 735 | low | 0.053 | Hospital-based | Ophthalmologist | 2003 |
| Shaffi and Bejiga [34] | 2005 | SNNP | Cs | 826 | low | 0.063 | Community-based | Not mentioned | 2003 |
| Demissie and Demissie [31] | 2014 | Oromia | Cs | 341 | low | 0.059 | Hospital-based | Optometrist | 2018 |
| Belete et al.[32] | 2016 | Amhara | Cs | 495 | low | 0.11 | School-based | Optometrist | 2016 |
| Gessese and Teshome [33] | 2020 | SNNP | Cs | 1271 | low | 0.072 | School-based | Optometrist | 2016 |
| Demissie and Solomon [30] | 2011 | Oromia | Cs | 112 | low | 0.053 | School-based | Ophthalmologist | 2010 |
| Asferaw et al. [29] | 2017 | Oromia | Cs | 104 | low | 0.019 | Community-based | Not mentioned | 2015 |
| Merrieet et al. [26] | 2021 | Amhara | Cs | 601 | low | 0.065 | Community-based | Optometrist | 2017 |

Cs = cross-sectional

size greater than or equal to 500, and six studies have a sample size less than 500. Nine studies were conducted in the Amhara region, seven in the Southern Nation and Nationality Region (SNNR), three in Addis Ababa, two in Tigray, and one in the Oromia region. About fifteen were school-based, five were community-based, and two were hospital-based studies. Concerning the risk of bias among studies, the majority (16) of studies were judged to have a low risk of bias [Table 1].

## The publication biases

The presence of publication bias was evaluated using funnel plots and Begg's tests at a significance level of less than 0.05. The findings revealed that publication bias was not significant for the studies on the prevalence of refractive error in Ethiopia (p = 0.096) [Fig 2].

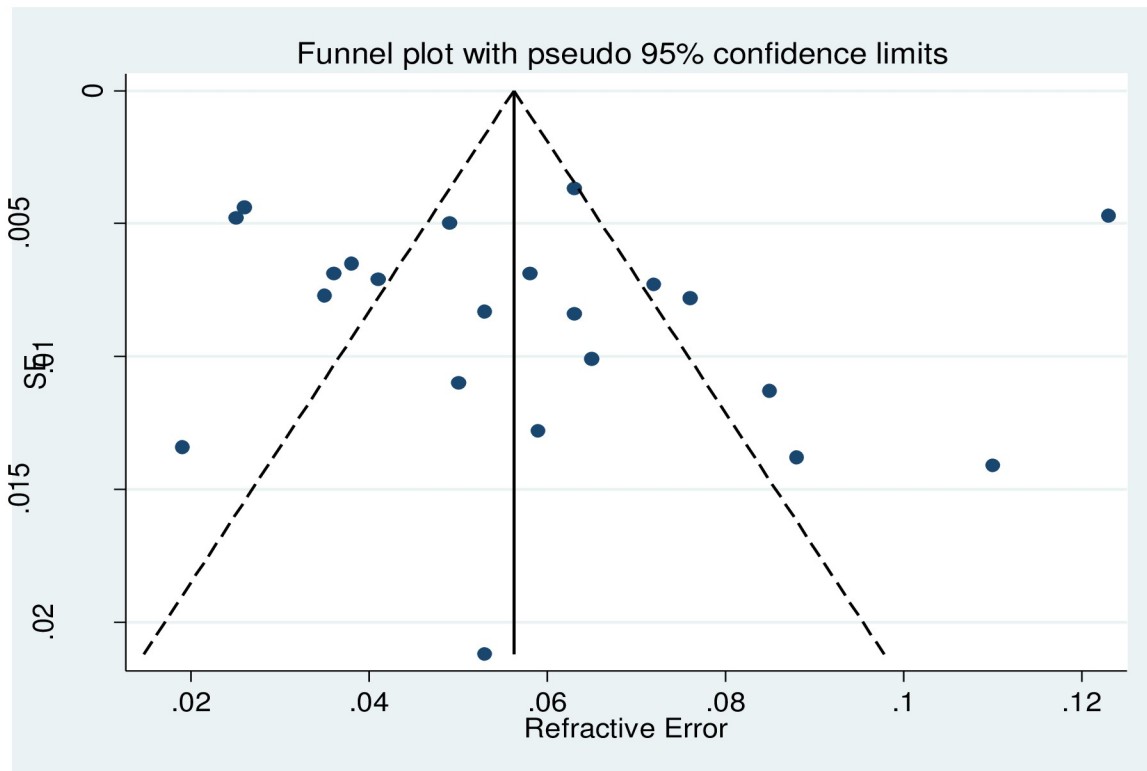

**Fig 2. Showing the publication bias status of studies included for meta-analysis on visual impairment due to refractive error in Ethiopia,2021.**

### Sensitivity analysis

The presence of a single study effect on the pooled prevalence of refractive error was tested using meta-influence analysis and revealed that there is no study significantly affecting the pooled prevalence of refractive error in Ethiopia [Fig 3].

### Prevalence of refractive error in Ethiopia

The pooled prevalence of refractive error in Ethiopia using the random effect model was estimated to be 6% (95% CI, 5–7) with a significant level of heterogeneity ($I^2$ = 94.4%; p < 0.001) [Fig 4].

The pooled prevalence of visual impairment due to refractive was analyzed by subtypes, and pooled prevalence was estimated to be 4% [Fig 5], 2% [Fig 6], and 1% [Fig 7] for myopia, astigmatism, and hyperopia, respectively. The heterogeneity was shown to be 94.6% for myopia, 60% for astigmatism, and 78.8% for hyperopia.

Subgroup analysis was conducted by region, year of publication, year of study, sample size, professional diagnosing refractive error, study setting, and risk bias of studies. The sub-group conducted by region identified a lower prevalence of refractive error among studies conducted in Addis Ababa (4%), and the heterogeneity was decreased to 0% among studies conducted in Addis Ababa and the Tigray region. The pooled prevalence of refractive error was not varied by the year of publication, but the heterogeneity was decreased to 60% among studies published from 2011 to 2015. Similarly, the pooled prevalence was not varied by the year of data

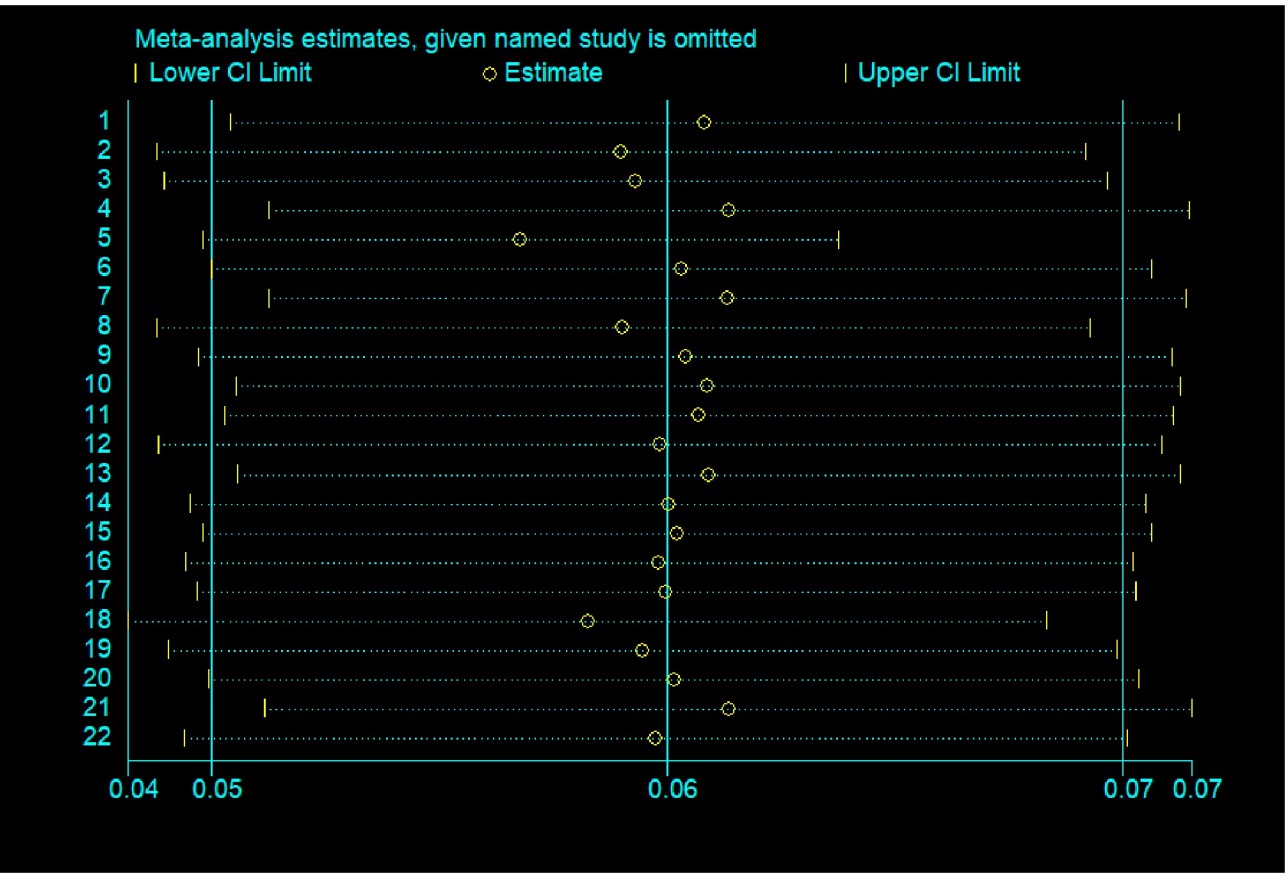

**Fig 3. Showing the sensitivity analysis of meta-analysis on visual impairment due to refractive error in Ethiopia, 2021.**

collection, but the heterogeneity is shown to decrease significantly (53%) among studies conducted between 2000 and 2010. Subgroup analysis conducted on the pooled prevalence of refractive error was highest among studies with a sample size between 500–750 (7%). The lowest result was identified among studies with a sample size greater than 750 (5%). In this subgroup, the heterogeneity varied from 67% among studies with a sample size between 751–1000 to 97.1% among studies with a sample size between 500–750. The pooled refractive error prevalence was 4% (95% CI: 3–6) among community-based studies. The heterogeneity decreased from 96% to 0% after subgroup analysis based on the study setting. The pooled prevalence of refractive error was identified to be 7% among studies where refractive error diagnosis was made by an optometrist and 4% among studies where an ophthalmologist made it. In this subgroup, the heterogeneity decreased to 68% among studies where an ophthalmologist made the diagnosis, while it was 95.2% among studies where the diagnosis was made by an optometrist [Table 2].

## Discussion

This meta-analysis is the first pooled prevalence of visual impairment due to refractive error in Ethiopia. The pooled prevalence of visual impairment due to refractive error was found to be 6% in Ethiopia, and myopia was identified as the commonest cause of visual impairment due

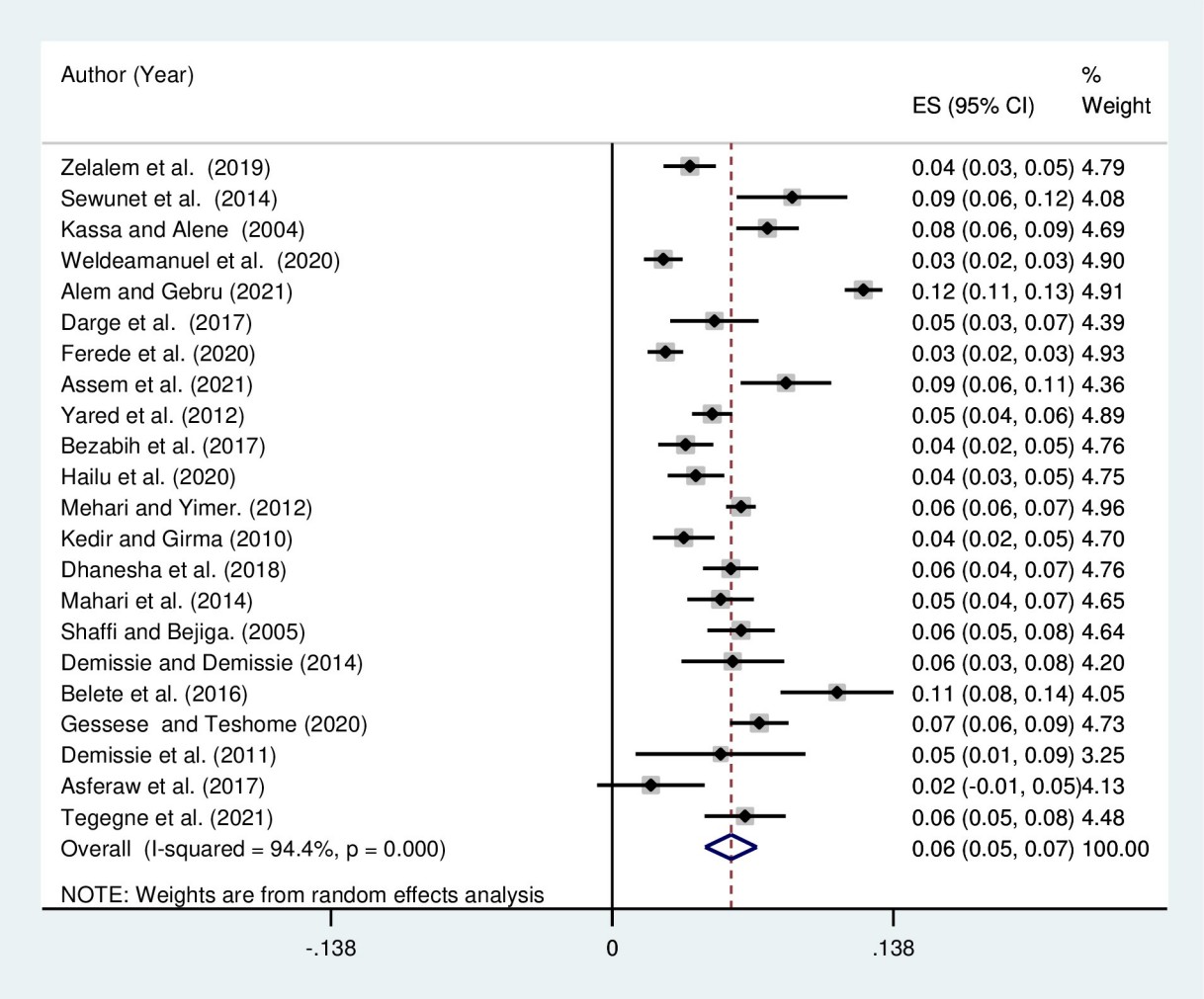

**Fig 4. Funnel plot showing the pooled prevalence of visual impairment due to refractive error in Ethiopia, 2021.**

to refractive error. This finding is higher than the pooled prevalence of visual impairment due to RE reported by a meta-analysis conducted in sub-Saharan Africa in 2015 [5]. It is also much higher than the prevalence of uncorrected RE reported by the global burden of disease (0.12%) in 2020 [41]. Further, the current finding is higher than the pooled prevalence of visual impairment due to RE reported by a global meta-analysis conducted in 2004 (1%) [42]. The difference might be explained by the variation of the study period of global metanalysis, which included studies conducted before 2004. The finding of this meta-analysis was also higher than other global meta-analyses conducted in 2010, which revealed the prevalence to be 2.1% [6]. The difference might be from sociodemographic and eye care service differences among various countries included in the analysis.

Subgroup analysis by region in this meta-analysis showed the lowest prevalence of refractive error among studies conducted in Addis Ababa. The difference among the regions may be due to variation in universal health coverage among different regions of Ethiopia. The universal health coverage was 52.2% in Addis Ababa, while 26% in the Oromia region [43].

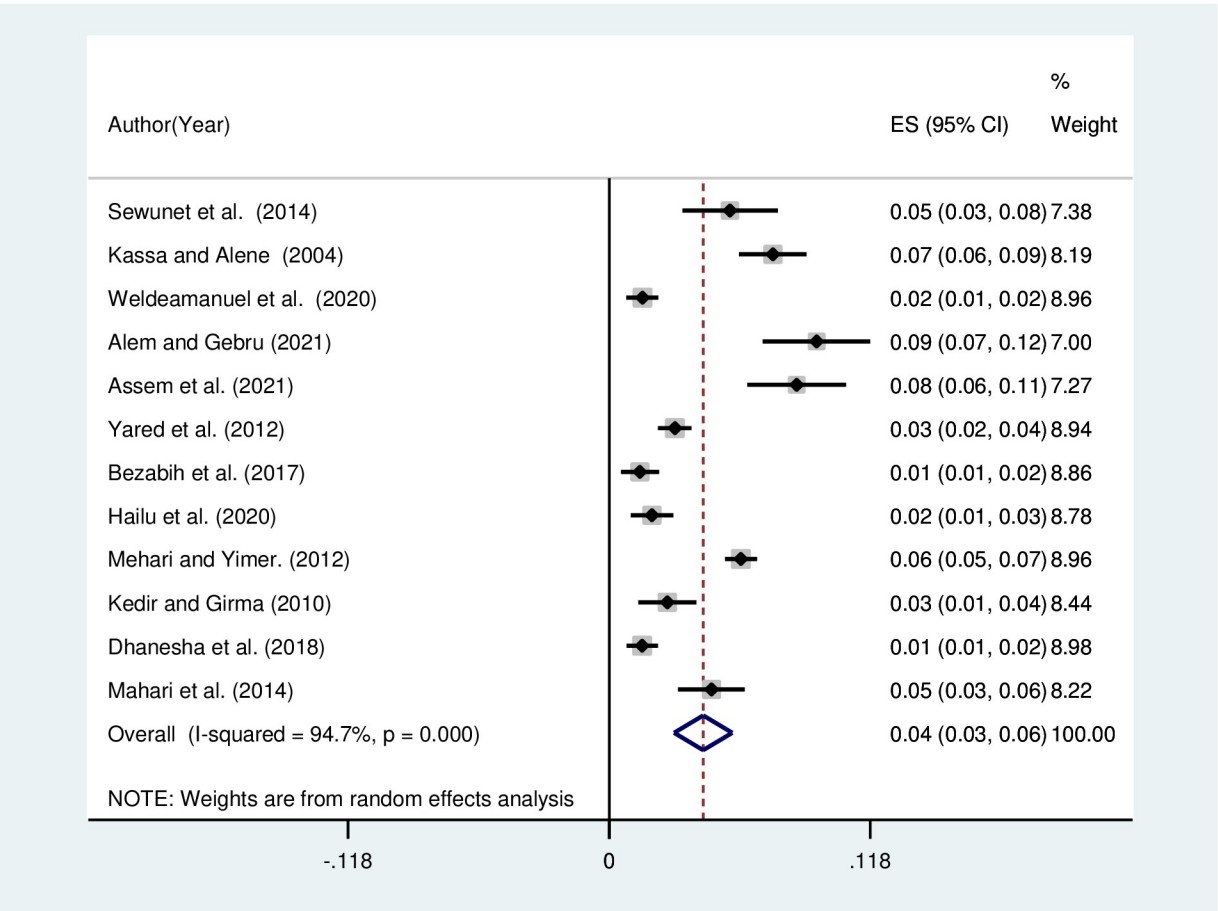

**Fig 5. Funnel plot showing the pooled prevalence of visual impairment due to myopia in Ethiopia, 2021.**

Subgroup analysis conducted on the pooled prevalence of visual impairment due to refractive error was highest among studies with a sample size between 500–750. At the same time, the lowest result was identified among studies with a sample size greater than 750. This difference may be because a small sample size has an overestimation effect compared with large sample size studies [44]. Further studies with a low sample size have a higher margin of error, and a large margin of error elongates the width of the confidence interval and gives an imprecise measure of the common effect size [45].

The pooled prevalence of visual impairment due to refractive error was higher among studies where an optometrist made a refractive error diagnosis than among studies where an ophthalmologist made a refractive error diagnosis. This difference can be explained by the difference in the skill of professionals in making an appropriate diagnosis of refractive errors. Some studies reported that diagnoses of ocular disease made by optometrists vary from those made by an ophthalmologist [46–49].

Generally, the current systematic review and meta-analysis reveal the prevalence of refractive error in Ethiopia, giving essential evidence for policymakers, clinicians, and other concerned entities who have previously overlooked the burden of refractive error on visual impairment.

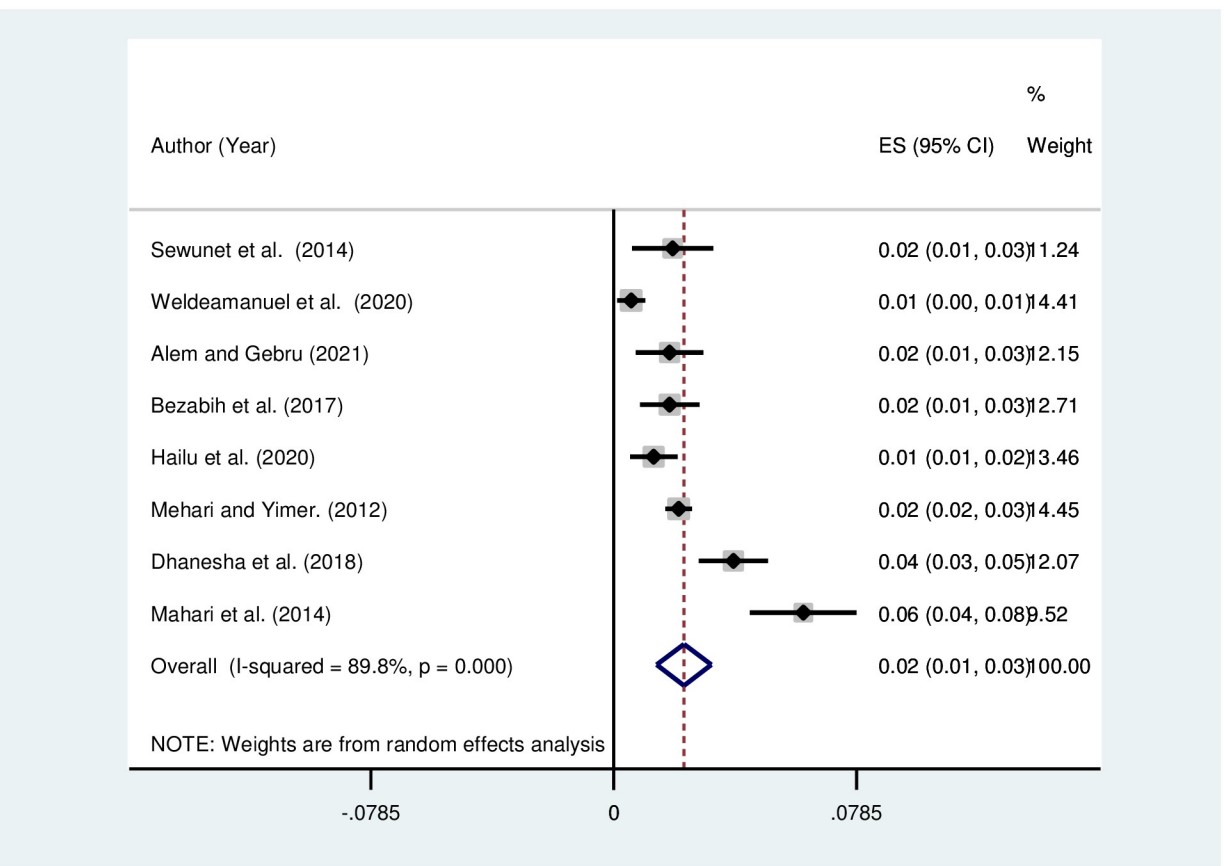

**Fig 6. Funnel plot showing the pooled prevalence of visual impairment due to astigmatism in Ethiopia, 2021.**

## Conclusion

In Ethiopia, the pooled prevalence of visual impairment due to refractive error was high. About one in twenty-five Ethiopian children and adolescents are affected by visual impairment due to myopia. Since the refractive error is a correctable cause of visual impairment, it is important to have a national figure in Ethiopia. Recognizing a high prevalence of visual impairment due to refractive error in Ethiopia may prompt policymakers to implement effective control and prevention initiatives to reduce the refractive error burden in Ethiopia.

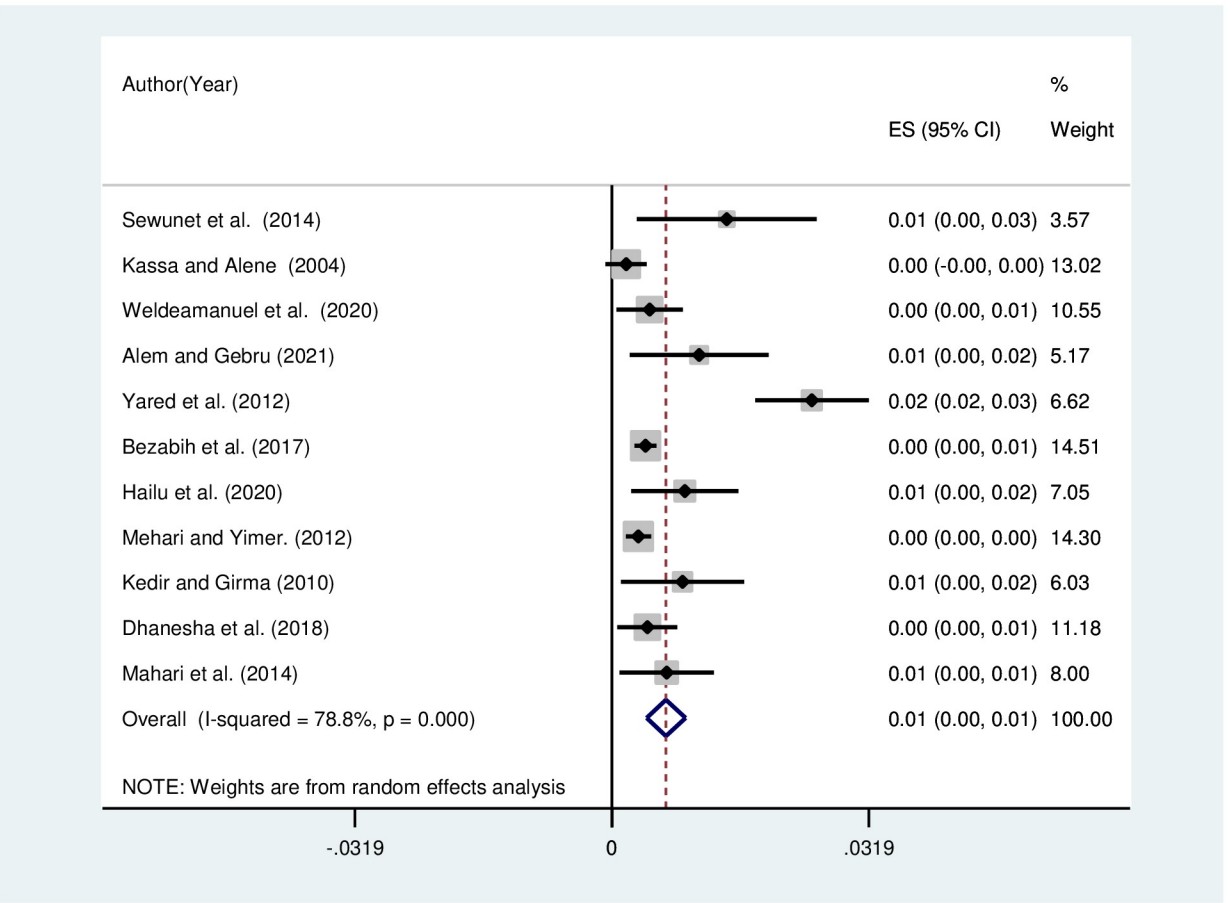

**Fig 7. Funnel plot showing the pooled prevalence of visual impairment due to hyperopia in Ethiopia, 2021.**

**Table 2. Sub-group analysis by region, year of publication, year of study, professional diagnosed RE, study setting, and sample size on the prevalence of visual impairment due to RE in Ethiopia, 2021.**

| Sub-group analysis | Number of studies | VI due to RE% | 95% confidence interval | Heterogeneity ($I^2$%) | P-value |
|---|---|---|---|---|---|
| **Sub-group analysis by year of publication** | | | | | |
| 1. < 2011 | 4 | 6.0 | 4.0–8.0 | 79.5 | P = 0.002 |
| 2. 2011–2015 | 5 | 6.0 | 5.0–7.0 | 60.6 | P = 0.038 |
| 3. 2016–2021 | 13 | 6.0 | 4.0–7.0 | 96.5 | P < 0.001 |
| **Sub-group analysis by year of study** | | | | | |
| 1. 2000–2010 | 6 | 6.0 | 4.0–8.0 | 95.8 | P< 0.001 |
| 2. 2011–2021 | 16 | 6.0 | 5.0–7.0 | 53.0 | P = 0.059 |
| **Sub-group analysis by professional who diagnosed RE** | | | | | |
| 1. Ophthalmologist | 6 | 4.0 | 3.0–5.0 | 68.0 | P = 0.008 |
| 2. Optometrist | 11 | 7.0 | 2.0–9.0 | 95.2 | P < 0.001 |
| **3. Not mentioned** | **5** | 5.0 | 5.0–6.0 | 82.1 | P < 0.001 |
| **Sub-group analysis by regions** | | | | | |
| 1. Amhara region | 8 | 6.0 | 5.0–6.0 | 91.4 | P < 0.001 |
| 2. Oromia | 3 | 4.0 | 2.0–7.0 | 59.8 | P < 0.083 |

*(Continued)*

**Table 2.**  (Continued)

| Sub-group analysis | Number of studies | VI due to RE% | 95% confidence interval | Heterogeneity ($I^2$%) | P-value |
|---|---|---|---|---|---|
| 3. Addis Ababa | 3 | 4.0 | 3.0–5.0 | 0.0 | P = 0.556 |
| 4. Tigray | 1 | 6.0 | 4.0–7.0 | - | - |
| 8. Southern Nation Nationality Region of Ethiopia | 7 | 6.0 | 3.0–9.0 | 97.5 | P < 0.001 |
| Sub-group analysis by study setting | | | | | |
| 1. Hospital-based | 2 | 5.0 | 4.0–7.0 | 0.0 | P < 0.069 |
| 2. School-based | 15 | 6.0 | 5.0–8.0 | 96.2 | P < 0.001 |
| 3. Community based | 5 | 4.0 | 3.0–6.0 | 73.5 | P < 0.004 |
| Sub-group analysis by sample size | | | | | |
| 1. <500 | 6 | 6.0 | 4.0–9.0 | 81.4 | P < 0.001 |
| 2. 501–750 | 6 | 7.0 | 3.0–10.0 | 97.1 | P < 0.001 |
| 3. 751–1000 | 3 | 5.0 | 3.0–6.0 | 67.0 | P = 0.048 |
| 4. 1001–1250 | 3 | 5.0 | 2.0–8.0 | 94.5 | P < 0.001 |
| 5. >1250 | 4 | 6.0 | 5.0–7.0 | 94.4 | P < 0.001 |

## Supporting information

**S1 Checklist. PRISMA checklist.**
(DOC)

**S1 File. Search terms used on different databases for visual impairment due to refractive error in Ethiopia.**
(DOCX)

**S1 Table. Showing the risk of bias assessment of included study for metanalysis of visual impairment in Ethiopia.**
(DOCX)

## Author Contributions

**Conceptualization:** Daniel Atlaw, Biniyam Sahiledengele, Sisay Degno, Ayele Mamo, Demisu Zenbaba, Habtamu Gezahegn, Fikreab Desta, Wogene Negash, Tesfaye Assefa, Mujib Abdela, Abbul Hasano, Gashaw Walle, Chala Kene, Degefa Gomora, Vijay Kumar Chattu.

**Data curation:** Daniel Atlaw, Zerihun Shiferaw, Biniyam Sahiledengele, Sisay Degno, Ayele Mamo, Demisu Zenbaba, Habtamu Gezahegn, Fikreab Desta, Tesfaye Assefa, Mujib Abdela, Abbul Hasano, Gashaw Walle, Chala Kene, Degefa Gomora, Vijay Kumar Chattu.

**Formal analysis:** Daniel Atlaw, Zerihun Shiferaw, Biniyam Sahiledengele, Sisay Degno, Ayele Mamo, Demisu Zenbaba, Habtamu Gezahegn, Fikreab Desta, Wogene Negash, Tesfaye Assefa, Mujib Abdela, Abbul Hasano, Gashaw Walle, Chala Kene, Degefa Gomora, Vijay Kumar Chattu.

**Funding acquisition:** Tesfaye Assefa.

**Investigation:** Daniel Atlaw, Biniyam Sahiledengele.

**Methodology:** Daniel Atlaw, Zerihun Shiferaw, Biniyam Sahiledengele, Sisay Degno, Habtamu Gezahegn, Fikreab Desta, Wogene Negash, Abbul Hasano, Gashaw Walle, Degefa Gomora, Vijay Kumar Chattu.

**Project administration:** Daniel Atlaw.

**Resources:** Daniel Atlaw, Sisay Degno, Abbul Hasano, Gashaw Walle, Chala Kene.

**Software:** Daniel Atlaw, Biniyam Sahiledengele, Sisay Degno, Abbul Hasano, Gashaw Walle.

**Supervision:** Daniel Atlaw, Zerihun Shiferaw, Ayele Mamo, Habtamu Gezahegn, Fikreab Desta, Mujib Abdela, Degefa Gomora.

**Validation:** Daniel Atlaw, Zerihun Shiferaw, Biniyam Sahiledengele, Ayele Mamo, Demisu Zenbaba, Habtamu Gezahegn, Fikreab Desta, Mujib Abdela, Abbul Hasano, Chala Kene, Degefa Gomora.

**Visualization:** Daniel Atlaw, Zerihun Shiferaw, Demisu Zenbaba, Fikreab Desta, Tesfaye Assefa, Mujib Abdela, Chala Kene, Degefa Gomora.

**Writing – original draft:** Daniel Atlaw, Sisay Degno, Demisu Zenbaba, Habtamu Gezahegn, Fikreab Desta, Wogene Negash, Tesfaye Assefa, Mujib Abdela, Abbul Hasano, Gashaw Walle.

**Writing – review & editing:** Daniel Atlaw, Zerihun Shiferaw, Biniyam Sahiledengele, Ayele Mamo, Chala Kene, Degefa Gomora, Vijay Kumar Chattu.

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
