## [Decision Letter · Decision Letter 0]

9 Jun 2022

PONE-D-22-10704Prevalence of Visual Impairment due to Refractive Error among children and adolescents in Ethiopia: A Systematic Review and Meta-Analysis. PLOS ONE

Dear Dr. Atlaw,

Thank you for submitting your manuscript to PLOS ONE. After careful consideration, we feel that it has merit but does not fully meet PLOS ONE’s publication criteria as it currently stands. Therefore, we invite you to submit a revised version of the manuscript that addresses the points raised during the review process.

We look forward to receiving your revised manuscript.

Kind regards,

Consolato M. Sergi

Academic Editor

PLOS ONE

Journal Requirements:

2. Please include your tables as part of your main manuscript and remove the individual files. Please note that supplementary tables (should remain/ be uploaded) as separate "supporting information" files

Additional Editor Comments:

Thank you for sending this manuscript to Plos One. After careful revision of the reviewers' concerns and criticisms, I would suggest that the authors need to address all issues from both reviewers in detail. I strongly suggest the authors revise the manuscript and replying point-by-point to the major and minor concerns.

Reviewers' comments:

Reviewer's Responses to Questions

**Comments to the Author**

1. Is the manuscript technically sound, and do the data support the conclusions?

Reviewer #1: Yes

Reviewer #2: Yes

2. Has the statistical analysis been performed appropriately and rigorously? 

Reviewer #1: Yes

Reviewer #2: Yes

3. Have the authors made all data underlying the findings in their manuscript fully available?

Reviewer #1: Yes

Reviewer #2: Yes

4. Is the manuscript presented in an intelligible fashion and written in standard English?

Reviewer #1: No

Reviewer #2: Yes

5. Review Comments to the Author

Reviewer #1: Firstly, the data provided does indeed support the conclusion, however more precision is needed on some points. For example, the text mentioned a global refractive error of 12%, but did not note the year in the text. The year is important as the authors mentioned that the prevalence of refractive error can change with better healthcare.

Few English mistakes have been highlighted in the attached review, with attached comments.

Reviewer #2: Topic:

Prevalence of Visual Impairment due to Refractive Error among children and adolescents in Ethiopia: A Systematic Review and Meta-Analysis.

The authors examined the pooled prevalence of visual impairment due to refractive error in Ethiopia. They found high value of the prevalence of visual impairment owing to refractive error. About one in twenty-five Ethiopian children and adolescents are affected by visual impairment due to refractive error.

Below are few suggestions:

Statistical methods and analysis section:

Under this section, the authors need to cite the relevant sources guiding their choices. That is, the articles that motivated the methods they opted for should be acknowledged with respect to heterogeneity, random effects, and publication bias assessments. For example, for publication bias assessment, the following article could be cited: Egger, M.; Davey Smith, G.; Schneider, M.; Minder, C. Bias in meta-analysis detected by a simple, graphical test. BMJ 1997, 315, 629–634.

This study is more about the assessment of prevalence. The influence of other factors on prevalence involving a meta-regression analysis should be treated on a different occasion. It is most likely a different topic, as it goes beyond the specific objective of the review, i.e. summary of the pooled prevalence of visual impairment due to refractive error. Hence, kindly remove the highlighted statement about meta-regression.

Further remarks:

Beware of the consistency in the spelling of the random-effects model throughout the article. It is ‘random-effects’ and not ‘random-effect’.

Table 2: Please add the sample size for each sub-group in this table. That is, you may mention the number of studies involved in each sub-group in a separate column.

Please see the full text for all the minor suggestions.

6. PLOS authors have the option to publish the peer review history of their article (what does this mean?). If published, this will include your full peer review and any attached files.

Reviewer #1: **Yes: **Youssef Mahfouz

Reviewer #2: **Yes: **Joseph F. Feulefack

---

## [Author Response · Author response to Decision Letter 0]

14 Jun 2022

To: PLOS ONE editorial

Response to reviewers

Additional Editor Comments:

Thank you for sending this manuscript to Plos One. After careful revision of the reviewers' concerns and criticisms, I would suggest that the authors need to address all issues from both reviewers in detail. I strongly suggest the authors revise the manuscript and replying point-by-point to the major and minor concerns.

Authors response -Dear respected editorial members of Plos One thank very much for your detailed and constructive comments and suggestions. We have replayed reviewers concern point by point and We have amended all journal requirement requested.

Reviewers' comments:

5. Review Comments to the Author

Reviewer #1: Firstly, the data provided does indeed support the conclusion, however more precision is needed on some points. For example, the text mentioned a global refractive error of 12%, but did not note the year in the text. The year is important as the authors mentioned that the prevalence of refractive error can change with better healthcare.

Author response -Dear respected reviewer #1; Thank you very much for your generous comment that has significantly improved our manuscript. We have added year of study for most of studies referenced in the introduction and discussion. For instance, see page 3 second paragraph line number 1,2 and 4 … page 10 discussion line number 5, 6.

Few English mistakes have been highlighted in the attached review, with attached comments.

Author response – Thank very much for detailed poofread we have corrected comments on the pdf as well as we have repeated proofread and the whole manuscript was amended once again. Any change made after proofreading was highlighted in red color on supplementary file.

New acronym, what is the delineation? Refractive error (RE)

This makes it a run-on sentence. The highlighted text should be a new sentence. Corrected please see page 3 last paragraph line number 2.

Is there a more recent study for comparison? - more recent reference was used for comparison of result in addition to previous reference. See page 10.

Reviewer #2: Topic:

Prevalence of Visual Impairment due to Refractive Error among children and adolescents in Ethiopia: A Systematic Review and Meta-Analysis.

The authors examined the pooled prevalence of visual impairment due to refractive error in Ethiopia. They found high value of the prevalence of visual impairment owing to refractive error. About one in twenty-five Ethiopian children and adolescents are affected by visual impairment due to refractive error.

Authors response – Dear respected reviewer #2 Thank you very much for your important comments and suggestions that advance our manuscript.

Below are few suggestions:

Statistical methods and analysis section:

Under this section, the authors need to cite the relevant sources guiding their choices. That is, the articles that motivated the methods they opted for should be acknowledged with respect to heterogeneity, random effects, and publication bias assessments. For example, for publication bias assessment, the following article could be cited: Egger, M.; Davey Smith, G.; Schneider, M.; Minder, C. Bias in meta-analysis detected by a simple, graphical test. BMJ 1997, 315, 629–634.

Author response- Thank you very much we have cited four references in the method section including the suggested reference see page 6.

37. Tufanaru C, Munn Z, Stephenson M, Aromataris E. Fixed or random effects meta-analysis? Common methodological issues in systematic reviews of effectiveness. Int J Evid Based Healthc. 2015;13(3):196–207. 

38. Borenstein M, Hedges L V., Higgins JPT, Rothstein HR. A basic introduction to fixed-effect and random-effects models for meta-analysis. Res Synth Methods. 2010;1(2):97–111. 

39. Egger M, Smith GD, Schneider M, Minder C. Bias in meta-analysis detected by a simple , graphical test. BMJ. 1997;(May 2014):629–34. 

40. Lin L, Chu H, Murad MH, Hong C, Qu Z, Cole SR, et al. Empirical Comparison of Publication Bias Tests in Meta-Analysis. J Gen Intern Med. 2018;33(8):1260–7.

This study is more about the assessment of prevalence. The influence of other factors on prevalence involving a meta-regression analysis should be treated on a different occasion. It is most likely a different topic, as it goes beyond the specific objective of the review, i.e. summary of the pooled prevalence of visual impairment due to refractive error. Hence, kindly remove the highlighted statement about meta-regression.

Author response – Thank you very much we have deleted statement about meta regression see page 6

Further remarks:

Beware of the consistency in the spelling of the random-effects model throughout the article. It is ‘random-effects’ and not ‘random-effect’.

Author response- Thank you very much We have corrected random-effect into random-effects

Table 2: Please add the sample size for each sub-group in this table. That is, you may mention the number of studies involved in each sub-group in a separate column.

Author response- Thank you very much we have added new column on table two indicating number of studies included. See page 10 table 2

Please see the full text for all the minor suggestions.

Author response- Thank you very much we have revised the whole manuscript and any change made were highlighted in red color

---

## [Decision Letter · Decision Letter 1]

29 Jun 2022

Prevalence of Visual Impairment due to Refractive Error among Children and Adolescents in Ethiopia: A Systematic Review and Meta-Analysis.

PONE-D-22-10704R1

Dear Dr. Atlaw,

We’re pleased to inform you that your manuscript has been judged scientifically suitable for publication and will be formally accepted for publication once it meets all outstanding technical requirements.

Kind regards,

Consolato M. Sergi

Academic Editor

PLOS ONE

Additional Editor Comments (optional):

Reviewers' comments:

Reviewer's Responses to Questions

**Comments to the Author**

1. If the authors have adequately addressed your comments raised in a previous round of review and you feel that this manuscript is now acceptable for publication, you may indicate that here to bypass the “Comments to the Author” section, enter your conflict of interest statement in the “Confidential to Editor” section, and submit your "Accept" recommendation.

Reviewer #1: All comments have been addressed

Reviewer #2: All comments have been addressed

2. Is the manuscript technically sound, and do the data support the conclusions?

Reviewer #1: Yes

Reviewer #2: Yes

3. Has the statistical analysis been performed appropriately and rigorously? 

Reviewer #1: Yes

Reviewer #2: Yes

4. Have the authors made all data underlying the findings in their manuscript fully available?

Reviewer #1: Yes

Reviewer #2: Yes

5. Is the manuscript presented in an intelligible fashion and written in standard English?

Reviewer #1: Yes

Reviewer #2: (No Response)

6. Review Comments to the Author

Reviewer #1: (No Response)

Reviewer #2: In the second paragraph of the Abstract, the correct spelling is ‘random-effects’.

7. PLOS authors have the option to publish the peer review history of their article (what does this mean?). If published, this will include your full peer review and any attached files.

Reviewer #1: **Yes: **Youssef Mahfouz

Reviewer #2: **Yes: **Joseph Florent Feulefack

---

## [Editor Report · Acceptance letter]

2 Aug 2022

PONE-D-22-10704R1 

Prevalence of visual impairment due to refractive error among children and adolescents in Ethiopia: a systematic review and meta-analysis. 

Dear Dr. Atlaw:

I'm pleased to inform you that your manuscript has been deemed suitable for publication in PLOS ONE. Congratulations! Your manuscript is now with our production department. 

Kind regards, 

on behalf of

Professor Consolato M. Sergi 

Academic Editor

PLOS ONE